# Metabolic Engineering of Shikimic Acid Biosynthesis Pathway for the Production of Shikimic Acid and Its Branched Products in Microorganisms: Advances and Prospects

**DOI:** 10.3390/molecules27154779

**Published:** 2022-07-26

**Authors:** Sijia Wu, Wenjuan Chen, Sujuan Lu, Hailing Zhang, Lianghong Yin

**Affiliations:** 1State Key Laboratory of Subtropical Silviculture, Zhejiang A&F University, Hangzhou 311300, China; sigawu-9@sjtu.edu.cn (S.W.); c18553789305@163.com (W.C.); 2Zhejiang Provincial Key Laboratory of Resources Protection and Innovation of Traditional Chinese Medicine, Zhejiang A&F University, Hangzhou 311300, China; 3Shandong Fengjin Biopharmaceuticals Co., Ltd., Yantai 264005, China; lvsujuan@fjbiopharma.com; 4College of Life Sciences, Yantai University, Yantai 264005, China

**Keywords:** shikimate pathway, shikimic acid, quinic acid, chlorogenic acid, gallic acid, pyrogallol, catechol, metabolic engineering, synthetic biology technology

## Abstract

The shikimate pathway is a necessary pathway for the synthesis of aromatic compounds. The intermediate products of the shikimate pathway and its branching pathway have promising properties in many fields, especially in the pharmaceutical industry. Many important compounds, such as shikimic acid, quinic acid, chlorogenic acid, gallic acid, pyrogallol, catechol and so on, can be synthesized by the shikimate pathway. Among them, shikimic acid is the key raw material for the synthesis of GS4104 (Tamiflu^®^), an inhibitor of neuraminidase against avian influenza virus. Quininic acid is an important intermediate for synthesis of a variety of raw chemical materials and drugs. Gallic acid and catechol receive widespread attention as pharmaceutical intermediates. It is one of the hotspots to accumulate many kinds of target products by rationally modifying the shikimate pathway and its branches in recombinant strains by means of metabolic engineering. This review considers the effects of classical metabolic engineering methods, such as central carbon metabolism (CCM) pathway modification, key enzyme gene modification, blocking the downstream pathway on the shikimate pathway, as well as several expansion pathways and metabolic engineering strategies of the shikimate pathway, and expounds the synthetic biology in recent years in the application of the shikimate pathway and the future development direction.

## 1. Introduction

The shikimic acid (SA) pathway is a common approach for the biosynthesis of aromatic amino acids (L-tryptophan, L-phenylalanine, and L-tyrosine) and other secondary metabolites in fungi, bacteria, plants, and certain *Apicomplexa* parasites [1,2,3]. This pathway is essential for the synthesis of not only aromatic amino acids, but also proteins, vitamins (vitamins E and K), and structural blocks of electron-carrier compounds, such as cofactors and quinones [4,5,6]. The SA pathway has different metabolic branches in different microorganisms. After years of research, this pathway has been widely studied and utilized in *Escherichia coli*: it has various branches that have been successfully integrated through genetic and metabolic engineering to broaden the metabolic network of this pathway [7]. In recent years, researchers have studied the enzymes involved in the SA pathway and identified their roles in microbial cell survival and presence in the clinically relevant spectrum of species. Consequently, these enzymes have been regarded as new targets of enzyme inhibitors, and extensive research has been conducted on them [8,9].

SA, quinic acid (QA), gallic acid (GA), pyrogallol, chlorogenic acid, and catechol are the products of the SA pathway and its branching pathways, and they are widely used as synthetic raw materials in the medical, chemical, food, and other industries [7,10,11,12]. Among them, SA is an intermediate of the SA pathway, and its structure is highly functionalized with a six-member carbocyclic ring and three asymmetrical centers [7,13]. Therefore, it is widely used in synthetic biology for the synthesis of different compounds with considerable biological and pharmaceutical activities [14,15]. The most remarkable use of SA is in the influenza antiviral drug, oseltamivir (Tamiflu^®^), as a synthetic material. Oseltamivir is a potent and selective inhibitor of influenza neuraminidase and can be effectively used, via oral treatment, for the prevention of viral influenza [16]. Furthermore, SA also plays an important role in the inhibition of the coagulation system, prevention of thrombosis, anti-inflammation, and analgesia [15]. As a result, the demand for SA is rapidly increasing and its market price is hiking. During the swine flu outbreak in 2009, Roche reported that Tamiflu^®^ sales amounted to USD 3.5 billion, with packaging being performed 400 million times a year, and SA price rose to 1000 USD/kg [15,17]. It takes 1.3 g of SA to produce 10 doses of antiviral drugs, which can treat only a single patient. It has been estimated that 520,000 kg of SA is required per year to meet Tamiflu^®^ production. Even so, this production capacity may not be sufficient for influenza pandemics, especially those caused by more pathogenic and infectious strains [5].

At present, the main source of SA supply is the fruit of Chinese star anise, whose raw materials are greatly affected by the cultivation conditions, resulting in a wipe gap between SA supply and commercial needs. Additionally, the multistep extraction and purification processes are expensive, thereby rendering meeting the growing global demand for Tamiflu^®^ challenging [5,11,15,18]. It is well known that chemical-based strategies can result in industrial environmental issues and low yields [7,19]. With the development of biotechnology, high-yielding strains have been directly screened for fermentation primarily via mutagenesis and random screening technology. Owing to the development of metabolic engineering and gene manipulation technology, it is possible to obtain more ideal engineering strains through the directional transformation of metabolic pathways. To overcome and solve the above-mentioned problems, well established metabolic and fermentation engineering methods (for instance, using glucose for SA production by altering endogenous metabolism and the fusion of all biosynthesis pathways), which can alter cellular properties, have been advocated as sustainable alternative approaches to meet the market demand for SA [7,19]. Significant achievements have been made in SA production via these methods [20]. The microbial metabolic engineering method is also of great significance for QA, GA, and catechol production, thereby effectively meeting the market demand for these products [21,22,23].

Metabolic engineering of the shikimate pathway and its branching pathways has greatly improved the yield of SA and its production and derivatives. The emerging of new strategies, such as synthetic biology, are showing great prospects in compensating and promoting the application of traditional metabolic engineering. This review provides deep insights into the synthesis pathway and regulation mechanism of SA, QA, GA, and catechol, and summarizes the metabolic engineering strategy applied in microorganisms. It also proposes the application and future prospects of synthetic biotechnology, combined with the systemic metabolic engineering strategy in the construction of microbial strains for high-yield SA production.

## 2. The SA Pathway and Its Regulatory Mechanism in *E. coli*

The SA pathway is an intermediary metabolic pathway, which is acknowledged as the basis for the generation of aromatic amino acids and their derivatives in plants and microorganisms, and SA is the intermediate of this pathway. Although the SA pathway has been extensively studied in *E. coli*, metabolic modifications of the pathway have only limitedly improved SA yield. On one hand, this pathway relies on the glycolysis pathway [24] and phosphopentose pathway (PPP) [25] to provide two starting substrates, including phosphoenol pyruvate (PEP) and erythrose-4-phosphate (E4P), respectively. On the other hand, this pathway includes many branching pathways (Figure 1), which need to be blocked to orient the carbon flow to SA.

The SA pathway was first identified in *E. coli*, which is the most commonly used bacterium in the metabolic engineering of this pathway and its branches. As shown in Figure 1, SA, QA, GA, and catechol biosynthesis in *E. coli* include the following steps. PEP and E4P are generated from the Embden–Meyerhof–Parnas pathway and PPP, respectively, and converted by 3-deoxy-D-arabinoheptose-7-phosphate (DAHP) synthase into DAHP. Subsequently, 3-dehydroquinate (DHQ) synthase (encoded by *aroB*) converts DAHP into DHQ, which is further converted by DHQ dehydrase (encoded by *aroD*) into 3-dehydroshikimate (DHS). NADPH-dependent shikimate dehydrogenase (encoded by *aroE*) reduces DHS to SA. Thereafter, shikimate kinase (encoded by *aroL* and *aroK*) catalyzes the production of shikimate-3-phosphate (S3P) from SA. S3P is further converted into 5-enolpyruvylshikimate-3-phosphate (EPSP), which is then converted by a series of enzymatic reactions, into aromatic amino acids. DHQ, the precursor compound of QA, is converted into QA by QA dehydrogenase (encoded by *ydiB*). Another intermediary product, DHS, is the substrate for catechol and GA production. Its production is catalyzed by DHS dehydrase and protocatechuic acid (PCA) decarboxylase, which are encoded by exogenous genes *aroZ* and *aroY*, respectively, and DHS dehydrase and hydroxybenzoate hydroxylase, which are encoded by exogenous genes *aroZ* and *pobA**, respectively.

Among these enzymes, shikimate kinase and DHQ synthase are the rate-limiting enzymes of the SA pathway in *E. coli* [26]. DAHP synthase, shikimate dehydrogenase, and DHQ synthase are the key enzymes of this pathway. Compared with DAHP synthase and shikimate kinase Ⅱ, which are transcriptionally regulated, DHQ synthase, DHQ dehydrase, and shikimate dehydrogenase are constitutively expressed in *E. coli*. As shown in Figure 1, there are three DAHP synthase isoenzymes, including AroG, AroF, and AroH in *E. coli*, which are separately inhibited by different amino acids (L-Phe, L-Tyr, and L-Trp, respectively) [7], thereby limiting the substrate inflow. Next to DAHP, DHQ synthase controls the metabolic flux. SA kinase is inhibited by SA feedback. Additionally, several SA transporters encoded by the *shiA* gene in *E. coli* are mainly ingested from the extracellular [27]. 

## 3. SA Production by Metabolic Engineering

SA, chemically known as 3,4, 5-trihydroxy-1-cyclohexen-1-carboxylic acid, with a molecular formula of C_7_H_10_O_5_, molecular weight of 174.15, melting point of 185–187 ℃, and an optical rotation of 180°, is a white acicular crystal with an octanoic acid smell. SA has three hydroxyl groups, one double bond and one carboxyl group in its molecular structure. It has chiral isomer and is easily soluble in water, with solubility of 18 g/100 mL in water, and is insoluble in chloroform, benzene and petroleum ether. SA is a kind of hydroxylated unsaturated acid derivative of cyclohexane, which widely exists in nature. It is the precursor of many chiral drugs, such as alkaloids, vitamins, aromatic amino acids and indole derivatives [28]. Modern pharmacological studies have shown that SA mainly has anti-tumor, anti-thrombosis and cerebral ischemia, anti-inflammatory, anti-virus, analgesic, treatment of alopecia and other pharmacological effects, which has important medical value [4,28,29,30,31,32]. Nevertheless, it is worth noting that SA can increase the risk of esophageal cancer and gastric cancer. In addition, SA can also control cyclin to promote breast cancer cell proliferation through miR-300 induced NF-κB-dependent pathways [33].

### 3.1. Engineering to Collect the Carbon Flux for SA Accumulation

It is difficult to accumulate SA, an intermediary metabolite of the aromatic amino acid pathway, in wild-type strains. Therefore, downstream biochemical reactions must be blocked or restricted to produce SA [7].

Davis and Mignioli observed for the first time that an *aroA* deficient *E. coli* strain could secrete S3P into the culture medium, and S3P could be converted into SA by heating or acidification [34]. This served as the foundation for the construction of the EPSP synthase-deficient strain used for SA production. Iomantas et al. constructed an *aroA*-deficient mutant *Bacillus subtilis* strain, thereby blocking the downstream pathway after S3P, and facilitating SA (1.1 g/L) accumulation without S3P. It is speculated that S3P was phosphorylated to SA by phosphatase [35].

Deletion of the *aroK* or *aroL* genes facilitates the construction of several microorganisms that retain weak shikimate kinase activity to limit downstream metabolism. A certain amount of aromatic amino acids and vitamins can be simultaneously synthesized to meet the growth needs of these mutants. In a previous study, the *aroL*-deficient *E. coli* W3110.shik1 strain yield phosphorus and carbon (0.056 and 0.023 mol/mol, respectively) under conditions of phosphorus and carbon limitation, respectively [36]. In another study, SA production in a mutant strain, obtained by inactivating the *aroK* gene via antisense RNA interference and gene deletion technology, was increased to 1850 mg/L, which is 2.69 times higher than that in *E. coli* DHPYA-T7 [37]. Similarly, the gene disruption method based on homologous recombination was performed to generate *Bacillus megaterium* ∆*aroK*, and the SA yield in this strain, in a bioreactor (10L), was increased to 6 g/L, 12 times higher than that in wild-type strains [38]. Interestingly, Lee et al. proposed a new way of not completely blocking the synthesis of aromatic amino acids, and significantly promoting the accumulation of SA. In this study, *aroK* and arL in *E. coli* were knocked out, and the expression of *aroK* gene was regulated by growth phase-dependent mechanisms (*rpsM* or *rrnBP1*). Meanwhile, *aroB*, *aroG**, *ppsA,* and *tktA* genes were overexpressed. The SA production of SK4/*rpsM* was 1.28 times higher than that of SK4/*pLac* without growth phase-dependent regulation [39].

The *E. coli* SP1.1/pKD12.112 (∆*aroL*, ∆*aroK*) strain, which showed no shikimate kinase activity, completely blocked the metabolic pathway of SA conversion into aromatic amino acids. The strain was cultured with additional aromatic amino acids and vitamins, and SA (27.3 g/L), along with QA (12.6 g/L), DHS (4.4 g/L), was produced. By-product accumulation adversely affected SA yield and its subsequent separation and purification [22].

To investigate the balance between SA and its by-products, the *E. coli* strain SP1.1/pSC5.214A was constructed, in which de novo SA synthesis was completely blocked. After fermentation in an SA-containing medium, the molar ratio of SA:QA:DHS was 1.1:1.0:0.70, providing the reversibility of the multi-step reaction in the SA pathway. The formation of QA and DHS was the result of the equilibrium of the initial SA that underwent microbial catalysis [13].

QA weakens downstream processing. Quinic/shikimate dehydrogenase (YdiB) in *E. coli* is a bifunctional enzyme, that shows 25% homology with shikimate dehydrogenase (AroE) [40]. In contrast, AroE is more likely to catalyze DHS conversion into SA. YdiB is more involved in catalytic QA production, since its affinity towards NAD is 10 times higher than that towards NADP [41,42]. Additionally, quinic/shikimate dehydrogenase, encoded by *qsuD* in *Corynebacterium glutamate*, catalyzes the first step in QA/SA catabolism and does not participate in the SA pathway, which is required for the subsequent biosynthesis of various aromatic compounds [41,42]. Therefore, a shortage in quinic/shikimate dehydrogenase can reduce the yield of the by-product QA and strengthen downstream metabolism. The *E. coli* PB12 derivative strain *ydiB* produced SA titers as high as 8.2 g/L, with an SA yield of 0.24 mol/mol, and QA titers of 0.38 g/L, which is 75% lower than that produced by the parental strain, in a fermenter (1 L). Moreover, the molar ratio of SA:QA decreased from 16.45% to 6.17% [42].

DHS accumulation was attributed to the feedback inhibition of SA dehydrogenase by SA. Knop et al. optimized the codon of *aroD·E*, the gene encoding the DHQ dehydratase and SA dehydrogenase duplex enzyme in *Nicotiana tabacum*, and its heterologous expression in *E. coli* JB4/pJB5.291 (∆*aroD*, ∆*aroE*), resulting in SA synthesis (34 g/L) with a 15% increase in yield after 66 h [13]. The molar ratio of SA:QA:DHS was 29:1.0:5.7. DHS accumulation was decreased, and its effect on QA was more significant, thereby indicating that substrate selection by the bifunctional enzyme is more inclined towards DHS [13]. In another study, *aroE* gene overexpression of the PB12.SA22 strain compensated for the feedback inhibition of shikimate dehydrogenase by SA. Compared with the PB12.SA21 strain, the PB12.SA22 strain showed increased SA titers (from 5.07 g/L to 7.05 g/L), increased SA yield (from 0.21 mol/mol to 0.29 mol/mol), and decreased DHS production (from 2.49 g/L to 1.46 g/L) [43]. Fortunately, since DHS can be easily converted into SA, many studies on metabolic engineering have been performed to improve SA, using high DHS titers [18].

### 3.2. Engineering to Expand the Metabolic Flux for Enhanced SA Production

Collecting the carbon flow into SA alone cannot meet the needs of industrial production. Therefore, the promotion of the metabolic flux in the shikimate pathway through genetic and metabolic engineering has become extremely critical.

One of the main principles in shikimate pathway designing is increasing the supply of key precursors (PEP and E4P). PEP plays an important role as a phosphate donor in the transmembrane transport of glucose [44]. When cells are exposed to a carbohydrate mixture, the phosphotransferase system (PTS) blocks the expression of catabolic genes and inhibits the activity of the non-PTS sugar transport system through carbon catabolic metabolite repression (CCR) [44]. When glucose is used as the sole carbon source, glucose uptake through the PTS consumes about 50% of the PEP, resulting in limited flux to the shikimate pathway [45]. At present, strategies to avoid PEP consumption mainly employ the heterologous expression of the *Zymomonas mobilis* Glf protein or recruit *E. coli* galactose permease (GalP) to replace PTS-mediated glucose transport [5,46,47]. Furthermore, the carbon storage regulator, CsrA, controls glycogen synthesis and inhibits gluconeogenesis by regulating pyruvate kinase, PEP carboxykinase, and PEP synthetase, whereas the CsrA antagonist, CsrB enhances PEP utilization when its expression is increased [48,49,50]. Kogure et al. substituted glucokinase (Glk1, Glk2 and Ppgk) and endogenous myo-inositol transporter (IOLT1) for the PTS system in *Corynebacterium glutamicum* to reduce PEP consumption and increase glucose intake. Overexpression of the glyceraldehyde 3-phosphate gene *gapA* reduces the excessive accumulation of intermediates in the EMP pathway, and they use growth-arrested cells to reduce by products and the carbon fluxes required for growth. Therefore, they obtained SA 141 g/L, 0.49 g/g glucose by microbial fermentation, the highest yield to date [51,52].

Licona-Cassani et al. reported, for the first time, that the *pykA*-deficient strain *B. subtilis* strain is better than the PTS-deficient *B. subtilis* strain [53]. PEP conversion into PYR in the non-PTS system depends only on the activities of pyruvate kinase (encoded by *pykF* and *pykA*) and PEP synthase (encoded by *ppsA*). When *ncgl2008* and *ncgl2809*, which encode pyruvate kinase in *C. glutamicum*, were inhibited by the CRISPRi system, SA production increased by 51% and 14%, respectively [54]. *ppsA* overexpression in *E. coli* SP1.1/pKD15.071B increased SA accumulation from 52 g/L to 66 g/L and glucose yield from 0.18 mol/mol to 0.23 mol/mol [46]. Additionally, Escalante et al. demonstrated, for the first time, that simultaneous PTS and *pykF* inactivation increased the production of SA and its aromatic precursors in *E. coli* [43]. Although the above-mentioned metabolic regulation can significantly increase the PEP pool and thus SA production, a balanced supply of the two precursors is essential for the introduction of metabolic flux into the shikimate pathway; *tktA* or *talA* is usually overexpressed to increase the supply of E4P precursors [44,51]. It has also been reported that the overexpression of the *zwf*Ⅰ gene (encoding 6-phosphate glucose dehydrogenase) increases the E4P pool, thus increasing the accumulation of SA, but the effect is not particularly significant [55].

A PTS-defective *E. coli* 1.1 PTS/pSC6.090B strain with *aroL* and *aroK* deletion and *aroF^Fbr^*, *aroE*, *glf*, *glk,* and *tktA* overexpression showed accumulated SA (87 g/L) with 36% yield in a fermenter (10L) with yeast extract [46].

Another challenge is the complex regulatory system of the shikimate pathway. Metabolic flux can be significantly enhanced by modifying key enzymes in the shikimate pathway. In a previous study, on metabolic flux analysis, different strains showed diverse responses to genetic manipulation of the *aro* genes involved in the shikimate pathway and the *tkt* and *pyk* genes involved in PEP/E4P production and consumption [56].

In *B. subtilis*, *aroA* (encoding DAHP synthase) and *aroD* (encoding SA dehydrogenase) co-expression doubled SA production (3.2 g/L), and the effects of *aroD* were more significant than those of *aroA*. Hence, *pyk* absence could increase the SA titer to 3.46 g/L, whereas *tkt* overexpression showed no effect on SA production in *B. subtilis*. These results are significantly different from those in *E. coli* [56].

In a double gene (*aroK* and *aroL*) knockout *E. coli* strain, *tkt* was first overexpressed by replacing the natural promoters of the *pps* and *csrB* genes and integrating additional gene copies. The key gene clusters of *aroG^fbr^*, *aroB*, a*roE*, and *tktA* were directly integrated into the host chromosome, and their copy numbers were increased by triclosan induction. SA production in the newly constructed *E. coli* SA 110 was 1.34 g/L, which is 8.9 times higher than that in the parent strain. Another copy of the *tktA* gene under the control of the *5Ptac* promoter was inserted into the chromosome of *E. coli* SA110. SA titer was further increased to 1.70 g/L, and a glucose yield of 0.25 mol/mol was obtained. More interestingly, *pntAB* or *nadK* overexpression was induced in the above-mentioned recombinant bacterial strain to improve NADPH availability. The final SA 116 strain produced 3.12 g/L of SA with a glucose yield of 0.33 mol/mol [48].

In addition, in the previously mentioned work of Lee et al., while regulating the growth phase-dependent expression of the *aroK* gene, heterologously expressed *DHQ-SDH* gene from the wooded plant *Populus trichocarpa* in *Escherichia coli*, the constructed SK5/pSK6 produced SA 5.33 g/L [39]. The *DHQ-SDH* enzyme is a bifunctional enzyme in plants with a substrate channel, so the intermediate DHS produced at one site can be directly captured by the next active site, without loss due to diffusion, and eliminating side reactions. Thus, the metabolic flux from DHQ to SA was expanded and the catalytic rate was increased.

### 3.3. Engineering for the Inhibition of SA Transport

A large amount of SA was accumulated through the above-mentioned metabolic engineering method by modifying the SA pathway and central carbon metabolic (CCM) pathways; however, the reuse of SA by microorganisms in the later fermentation stage not only reduced the yield of SA, but also promoted by-product formation [7].

When the first three steps of the shikimate pathway in *E*. *coli* were blocked, the shikimate transporter protein (encoded by *shiA*) could absorb the exogenous SA to synthesize the aromatic amino acids and vitamins needed for bacterial growth [27,56]. This indicated that the *shiA* gene knockout can prevent the reingestion of SA by the cells in the medium and guide the balance in the direction towards the by-products. Therefore, a DAHP synthetase activity- and *shiA*-deficient *E. coli* sp1.1*shiA*/pSC5.214A strain was constructed. QA and DHS were detected in the SA-supplemented medium even after culture. This newly constructed strain, compared with the original strain, showed significantly decreased synthesis rate and by-product content. This phenomenon also indicated that there are other SA transport systems in *E. coli*, but the transport mechanism remains unclear [13,57].

CCR refers to the inhibitory effect of a fast metabolic carbon source on gene expression or protein activity involved in the metabolism of other carbon sources. Glucose is usually utilized by bacteria in environments where multiple carbon sources coexist [58]. Draths et al. reported that SA reuptake could be inhibited by increasing glucose utilization, thereby reducing QA formation [22]. The results indicated that SA (20.2 g/L), DHS (4.6 g/L), and QA (1.9 g/L) were synthesized in *E. coli* SP1.1/pKD12.112 under high-glucose utilization conditions. Similarly, SA (27.2 g/L), DHS (4.4 g/L), and QA (12.6 g/L) were synthesized under low-glucose utilization conditions. Apparently, maintaining high glucose concentration can dramatically decrease accumulation of the by-product, QA; however, yield of the main product was also decreased by 25% [58]. This might be due to the abundant glucose supply, thereby increasing by-product (acetic acid) production, inhibiting *E. coli* activity, and reducing SA yield. To solve the above-mentioned problems, Knop et al., in addition to transforming the PTS, supplemented the fermentation medium with the glucose structure analogue, methyl-α-D-glucopyranoside, and cultured the SA-producing strain *E. coli* SP1.1/pKD12.138 strain under glucose-limited conditions for 48 h to investigate the regulation of the shikimate transport system by CCR. SA was synthesized at 35 g/L with a 19% yield of, whereas the by-product, QA, was synthesized at only 2.8 g/L [13]. Hence, the addition of a glucose mimic was an obviously effective way to reduce by-product synthesis.

### 3.4. Strain Improvement Using Synthetic Biology Technology

The classic metabolic engineering strategy is quite mature, but it can also reach a bottleneck stage. Reconfiguration of biochemical networks by manipulating the microbial genetic code often introduces flux imbalances [59]. The feedback inhibition of upstream enzymes and the formation of by-products often result in unexpected SA accumulation. The genes encoding the SA pathway enzymes are not continuous in the genome and are differently regulated, thereby causing additional difficulties in genetic manipulation and the metabolic engineering of the SA pathway; hence, gene knockout may not be the best choice for genes involved in important metabolic pathways [60]. In recent years, the development of synthetic biology technology has allowed researchers to gain an in-depth understanding of the global nature of various biological elements in the SA pathway [5].

The development of synthetic biology has brought new concepts to the design and construction of genetic modules or metabolic engineering of biological processes. Various enzyme elements have been used as parts of genetic modules [60]. Promoter libraries and ribosomal-binding site (RBS) variants may be applied to explore a wider dynamic range of gene expression [59]. According to the literature, some researchers successfully constructed RBS libraries tailored to *aroG*, *aroB*, *aroD*, and *aroE*, thereby building continuous genetic modules [ribozyme insulator (RiboJ), transcription promoter (Ptac), terminator, and the *aroG*, *aroB*, *aroD*, and *aroE* genes] regulated by the same promoter (Ptac) in the RBS libraries applied in *C. glutamicum* RES167Δ*aroK*. The results showed that SA products by *C. glutamicum* RES167Δ*arok* strain with a genetic module (4.3 g/L) were 54 times higher than that by the non-genetic module strain (80 mg/L), during fermentation in flasks (250 mL) [48]. Chen et al. gradually improved the SA pathway through the modular expression of three key genes, including *aroG^fbr^*, *ppsA,* and *tktA,* using a batch-feeding process. SA (14.6 g/L) was finally produced in a bioreactor (7 L) with a glucose yield of 0.29 g/g [61].

In a previous study, the SA pathway was enhanced by regulating the intensity of different RBSs [60]. At present, the emergence of the CRISPRi system provides us with new, efficient, and time-saving means of gene expression regulation. By overexpressing *ncgl1512* (*tkt*) and downregulating *ncgl2008* (*pyk*), *ncgl2809* (*pyk*), and n*cgl1856* (transcriptional regulator of sugar metabolism) expression in CRISPR-Cas9 system, SA titer in a shake flask (250 mL) and fermentor (5 L) reached 7.76 g/L and 23.8 g/L, respectively, and this is the highest SA yield reported in *C. glutamicum* [54].

Maintaining the carbon flux balance between microbial cell growth and product production is a long-standing problem in the field of industrial microbes, while uncoupling between cell growth and product production requires precise manipulation of carbon flux, which can be achieved by regulating the expression of enzymes in various pathways and expression of the enzyme. The dynamic adjustment tool represents a new frontier, which can control the allocation of resources between competitive routes and direct carbon fluxes to the target pathways. Hou et al. constructed a dynamic bifunctional switch using a growth-coupled promoter and a degradation tag. In this study, the growth-coupled promoter can regulate the target gene at the transcriptional level, while *ssrA* degradation tags dependent on the ClpXP system can regulate the target gene at the post-translation level. This bifunctional molecular switch was used to separate *E. coli* cell growth from shikimic acid synthesis, resulting in 14.33 g/L SA [62]. In addition, a new study also shows a breakthrough in dual-function optical genetic switches. The researchers reprogrammed the widely used inhibitor TetR and tobacco etch virus protease TEVp to construct two optogenetic systems (TPRS and TPAS) to expand the current optogenetics toolkit, and by combining the two systems to construct a bifunctional photogenetic switch for orthogonal regulation of gene transcription and protein accumulation. When this bifunctional switch is applied to the biosynthesis of SA, SA of 35 g/L is produced in the minimum glucose medium without adding any chemical inducers and expensive aromatic amino acids, which is the highest titer reported by *E. coli* so far. When fermented with a rich culture medium, the titer was further increased to 76 g/L, which provides a promising method for the construction of economically attractive microbial cell factories [63]. Obviously, the dynamic switching module can solve the problem of carbon flux imbalance in traditional metabolic engineering.

In short, synthetic biology technology constructs all kinds of chassis cells from a modular point of view, combines biological components from different sources, and then designs a controllable artificial biological system, which is more in line with people’s expectations. It also breaks the limitations of the original natural evolution, under the premise that the overall coordination of microbial cells and the economic management principle of maintaining livelihood remains unchanged, and makes the cell economy run to a larger production pattern.

### 3.5. QA Production by Metabolic Engineering

QA is a cyclic polyhydroxy compound with optical activity. It is commonly found in jinna bark, tobacco leaves, carrot leaves, apples, peaches, coffee seeds, and a few in fungi and bacteria [23]. In addition to synthetic neuraminidase inhibitors GS401 and GS4104 [64], QA is widely used for the synthesis of anti-neoplastic agents (esperamicin-A), immunosuppressants (FK-506), anti-hepatits B drugs (BE-5), and chlorogenic acid, which are used for the prevention of cardiovascular disease, cancer, and the side effects of chemotherapy [65,66]. Furthermore, QA can be used as a food additive, a cosolvent, and an optical material. As the scope of QA action expands, new QA functions, such as radiation protection, antioxidant activity, anti-neuroinflammation, and anti-thrombosis, are discovered. Recent studies show that QA can inhibit vascular inflammation, such as in arteriosclerosis [67], and act as a promising antibiofilm agent to control *Staphylococcus aureus* formation. Hence, there is a growing demand for QA [44].

At present, the main methods of QA acquisition are plant extraction, chemical synthesis, enzyme engineering, and microbial fermentation. Among them, plant extraction results in low yield. Additionally, it is difficult to obtain an optically-active QA by chemical synthesis, and the process is complex and causes environmental pollution. Furthermore, enzyme engineering results in high yield, but is expensive [23], and microbial fermentation requires economic and environmental protection.

As shown in Figure 2, DHQ is a precursor compound of QA and an intermediary metabolite in the SA pathway. Therefore, to accumulate the target product QA, obtained mainly from the transformation of the CCM pathway, the downstream pathway must be blocked, and related genes must be overexpressed. The following metabolic strategies used for QA production are similar to those used for SA accumulation.

### 3.6. Regulation of the CCM Pathway

The CCM pathway is reformed to direct the carbon flow to DAHP synthesis, which is the key step the yield. On one hand, in studies conducted by Jia et al. and Patnaik et al., availability of the substrate E4P in the shikimate pathway was increased by transketolase Ⅰ (encoded by *tktA*) or transaldolase (encoded by *talB*) overexpression [68,69]. On the other hand, Kim et al. overexpressed the *tktA* and *aroF^fbr^* genes to increase DAHP secretion in *E. coli* by 40 times [70], and knocked out the *pykF* and *pykA* genes *pykA* (encoding pyruvate kinase) overexpressed *ppsA* (encoding PEP synthase) by modifying the PTS to increase the availability of the substrate, PEP. After analyzing the CCM flux, Meza et al. found that inactivation of either of the above-mentioned processes would increase PEP availability in the PTS^−^glc^+^ strain [6]. Studies show that simultaneously deleting two *pyk* genes can increase the DAHP yield by three times [71].

### 3.7. Modification of the Shikimate Pathway

The shikimate pathway is a metabolic tree with many branches. After introducing the carbon flow into the shikimate pathway, the QA biosynthesis pathway must be constructed. Above all, blocking the downstream pathway, by knocking out the *aroD* gene and blocking the flow of DHQ to DHS, is necessary for QA accumulation. Additionally, selecting the appropriate expression vector or introducing a strong promoter to overexpress the related enzyme (DAHP synthetase, QA dehydrogenase, or 3-dehydroquinic acid synthetase)-encoding genes is aimed at directing more carbon sources to QA [23]. An *E. coli* strain of AB2848*aroD*/pKD136/pTW8090A, which was modified by metabolic engineering as mentioned above, was cultured continuously for 24 h to synthesize QA (10.7 g/L) [10]. In previous studies, quinine dehydrogenase was not found in *E. coli*, but it was heterogeneously introduced in *Aspergillus nidulans, Neurospora crassa*, *Klebsiella pneumoniae*, and other microorganisms via expression of the *qutB* gene [10,23]. In recent years, researchers have found that *E. coli*, *Salmonella typhimurium*, *Streptococcus pneumoniae,* and *Haemophilus influenzae* possess dual specificity quinate/shikimate NAD-dependent dehydrogenase (encoded by *ydiB*) [41]. In a study by García et al., *ydiB* in *E. coli* was overexpressed, thereby increasing the molar yield of QA by 500% [42].

As shown in Figure 2, chlorogenic acid is a combination of caffeic acid and QA [72]. It is rich in coffee, fruits, and vegetables and is the main source of caffeic acid in the human diet. Furthermore, chlorogenic acid prevents cardiovascular disease, cancer, and the side effects of chemotherapy; thus, it is one of the current research hotspots [65]. In *E. coli*, most of the DHQ is converted into DHS, and the remainder is converted into QA. An *aroD* mutant *E*. *coli* B-101strain showed DHQ accumulation, but synthesized caffeoyl-3-dehydroquinate instead of chlorogenic acid. However, a large amount of QA was accumulated by overexpressing the *ydiB* gene in the B-101 strain, thereby sharply increasing chlorogenic acid production to 450 mg/L [72].

## 4. GA and Pyrogallol Production by Metabolic Engineering

GA, chemically named 3, 4, 5-trihydroxybenzoic acid, is a polyphenolic compound found in tea, nuts, apple peels, green tea, oak bark, and grapes [73,74]. Like SA, GA contains a versatile six-carbon ring structure and three asymmetrical chiral carbon atom centers, and potential pharmacodynamic derivatives can be obtained from it by chemical modification [75,76]. Recent studies show that GA has antibacterial, anti-inflammatory, antiviral, antitumor, and antioxidant properties [73] and is used in the treatment of cardiovascular and neurodegenerative diseases [77,78,79]. As a natural product, GA can only be obtained by extraction from natural resources. Therefore, biosynthesis must be performed to obtain GA effectively [22].

Pyrogallol (1,2,3-benzenetriol) is one of the most important derivatives of GA. It is characterized by rapid self-oxidation, especially in alkaline solutions, and is, hence, widely used in gas deoxidation. The pyrophenol autoxidation method, which has been used to detect superoxide dismutase and other antioxidants and measure superoxide-scavenging activity to date, is based on this characteristic [80,81]. Additionally, pyrogallol is widely used in the food, leather, agricultural, electronic, paint, and pharmaceutical industries to synthesize bioactive compounds [82]. It can also regulate the expression of pro-inflammatory genes in bronchial epithelial cells to achieve anti-inflammatory effects [83].

### 4.1. Metabolic Engineering to Broaden the Shikimate Pathway for GA Production

GA, but not DHS dehydrogenase-mediated catalytic oxidation of DHS and PCA hydroxylase-mediated catalytic hydroxylation, was detected in the culture media with *Pseudomonas fluorescens*, *Phycomyces blakesleeanus*, *Aspergillus terreus*, *Enterobacter cloacae*, and recombinant *E. coli*; thus, there is no natural pathway for direct DHS conversion into GA in microorganisms. As shown in Figure 3, effective GA synthesis was achieved via an artificial biosynthesis pathway [43,84].

*E. coli* KL7 is an *aroE* mutant with exogenous insertion of the *aroZ* and *aroB* genes. It facilitates DHS accumulation and can be dehydrated by DHS dehydratase (encoded by *aroZ*) obtained from *K. pneumoniae* to PCA. When *pobA** and *aroF^fbr^* on the pSK6.161 plasmid were expressed in KL7 strain, PCA was hydroxylated into GA by p-hydroxybenzoate hydroxylase (encoded by *pobA**) obtained from *Pseudomonas aeruginosa*, and GA (20 g/L) was synthesized from glucose, with 12% yield [43,84]. In another study, the *pobA** gene was overexpressed in the directed mutant *Y385F/T294A* strain to improve p-hydroxybenzoate hydroxylase activity and the accumulated GA levels (1149.59 mg/L) [85].

### 4.2. Metabolic Engineering to Broaden the Shikimate Pathway for Pyrogallol Production

Theoretically, GA can be converted into pyrogallol by PCA decarboxylase (encoded by *aroY*), but the introduced PCA decarboxylase tends to decarboxylate intermediate PCA to catechol; thus, further pyrogallol synthesis via the above-mentioned pathway remains unsuccessful [84]. Developing a new and efficient alternative artificial pathway for pyrogallol synthesis is challenging.

The structure of pyrogallol is very similar to that of the natural intermediate 2,3-dihydroxybenzoic acid (2,3-DHBA) of the shikimate pathway. The decarboxylation of the C1-position carboxyl group to a hydroxyl group in 2,3-DHBA may be effective for pyrogallol formation. Wang et al. identified the highly efficient 2,3-DHBA 1-monooxygenase and established an artificial pathway for de novo pyrogallol synthesis in *E. coli*, based on the structural similarity between the substrate and product (Figure 3). Further optimization, such as increasing the carbon flux, modularization, and slowing down pyrogallol self-oxidation, increased pyrogallol titer to 1035.75 mg/L in the shake flask test [82].

As shown in Figure 3, heterologous expression of *Y385F/T294A* PobA (from *Pseudomonas aeruginosa*) and PDC (from *Klebsiella pneumoniae subsp. Pneumoniae*) in *E. coli* could construct the artificial synthetic pathway of pyrogallol, but its titer did not significantly improve. Huo Y X fused CipA protein at the N-terminal of *Y385F/T294A* PobA to assemble protein crystalline inclusions (PCLs) and to improve the hydroxylation ability of 3, 4-DHBA. At the same time, the concentration of yeast extract was optimized, so that the SA titer of strain RHY5 reached 80 mg/L after 36 h of fermentation. The addition of signal peptide at the N-terminal of PDC can reduce the by-product catechol [86].

## 5. Catechol Production by Metabolic Engineering

Catechol (1, 2-dihydroxybenzene) is an aromatic compound that is soluble in water, ethanol, benzene, chloroform, and pyridine. As a fine chemical raw material, it is widely used in medicine (berberine and isoproterenol), pesticides (propoxur and carbofuran), spices, dyestuff, and rubber production [87,88,89].

The main methods used to produce catechol are phenol and m-diisopropylbenzene oxidation and coal tar distillation [21]. However, due to the complex chemical synthesis process, serious pollution, requirement of expensive catalysts, and other such problems, these methods have been phased out. Moreover, several *Pseudomonas* bacteria can convert different aromatic substrates into catechol or PCA. For example, using transposon Tn916 or adding tetracycline in a *Bacillus stearothermophilus* BR219 strain, which can degrade phenol during the stable phase, resulted in the inactivation of the gene that encoded catechol 2,3-dioxygenase involved in the phenol meta pathway, leading to catechol accumulation [90]. In another malodorous *Pseudomonas putida* strain that expressed toluene/benzene dioxygenase and lacked catechol 1,2-oxygenase and catechol 2,3-oxygenase activity, benzene was oxidized to produce catechol [91]. Nevertheless, the raw material of synthetic aromatic substrates is a non-renewable carbon source [92]. Consequently, biosynthesis of renewable carbon sources has become a hot topic. Since DHS is a substrate of catechol, the main method for catechol biosynthesis is the reasonable modification of the SA pathway.

### 5.1. The Metabolic Pathway of Catechol Production from DHS

Catechol is mostly produced via chemical synthesis using raw materials extracted from petroleum. The ability to produce catechol naturally is observed only in a small number of microbial species, and acquiring the required carbon source is a complex process; hence, the requirements of industrial production are not met [21]. Fortunately, with the further development of metabolic engineering, recombinant bacteria have been constructed for the production of catechol from simple carbon sources [93].

*E. coli*, a model organism, is the most commonly used bacterium in metabolic engineering. Draths and Frost identified an induction pathway for the conversion of D-glucose into catechol [92]. Based on metabolic engineering (Figure 4), the *E. coli* mutant AB2834 strain without shikimate dehydrogenase activity facilitated DHS accumulation and expressed the *aroZ* and *aroY* heterologous genes (encoding DHS dehydrogenase and protocatechuate decarboxylase, respectively) obtained from the *K. pneumoniae* A170-40 strain [92,93], resulting in a 33% a yield of catechol from glucose [93]. Additionally, catechol concentration in the culture medium was reduced by in situ resin-based extraction during the synthesis of catechol from glucose in *E. coli* WN1/pWL1.290A to reduce the microbial toxicity of catechol and increase the yield of catechol by 2% [94].

### 5.2. The Metabolic Pathway of Catechol Production from 4-HBA

To construct the pathway of catechol synthesis based on DHS, the *aroE* gene needs to be knocked out in order to make the carbon flow converge to catechol, but it also means that bacteria cannot synthesize aromatic amino acids and some vitamins independently, and need an extra nutrition element to meet the demand of bacteria growth itself [95]. As a result, this high cost is bad for industrialized production.

In order to solve the above problems, Pughs et al. constructed an artificial synthetic pathway with 4-HBA as the endogenous precursor in *E. coli*. As shown in Figure 4, only *ubiC* (encoding native choroidal pyruvate lyase) from *E. coli* bw25113, *pobA* (encoding 4-HBA hydroxylase) from *P. aeruginosa*, and *ECL* (encoding PCA decarboxylase) from *Enterobacter cloacae* ATCC 13,047 were co-expressed. At the same time, the strain can also produce its own aromatic amino acids and vitamins. Furthermore, because L-Phe is largely separated from chorismate, the *pheA* gene-encoding chorismate mutase/precursor dehydrogenase was knocked out, and the *E. coli* N74dpheA pUbic-PobA pECL constructed by the above engineering could produce 630 mg/L catechol in a batch bioreactor after 86 h fermentation, with a yield of 36.2 mg/g glucose [96].

### 5.3. The Metabolic Pathway of Catechol Production from Anthranilate (ANT)

The *P. aeruginosa* PAO1 strain can use ANT as a carbon and energy source [97]. This strain contains specific enzymes that can convert ANT into catechol through o-dihydroxylation, spontaneous deamination, and decarboxylation. Thus, as shown in Figure 4, amplifying the metabolic network of catechol in *E. coli* by metabolic engineering can be employed as another strategy for catechol production [21].

In one study, *trpD* gene mutation in *E. coli* strain W3110*trpD9923* with a tryptophan operon was transformed using the pJLB*aroG^fbr^tktA* plasmid to increase the carbon flow to the SA pathway. A recombinant bacterial system was constructed to produce ANT (14 g/L) in a glucose-fed batch culture. Subsequently, the *antABC* gene [encoding ANT 1,2-dioxygenase (AntDO)] in *P. aeruginosa* PAO1 was heterogeneously expressed in the transformed pTrc-*ant3* plasmid, and a new pathway for catechol production was reconstructed. Catechol titer in a bioreactor culture was 4.47 g/L with 16% yield [21].

The process of production of catechol from glucose can be used to design and improve the process of biosynthesis of organic compounds derived from catechol. For example, catechol can be completely converted into muconic acid (389 mg/L) by expressing heterologous AntDO and catechol 1,2-dioxygenase [98].

## 6. Conclusions and Future Directions

The SA pathway is a common pathway for aromatic amino acid synthesis and widely exists in plants and microorganisms. The important intermediary products in this pathway are of great value. Besides exhibiting extensive medicinal value, SA is the raw material for the synthesis of various alkaloids, aromatic amino acids, indole derivatives and chiral drugs; 3-dehydroshikimic acid exerts great antioxidant effects and is the raw material for the synthesis of catechol, GA, and other important chemical products; and QA derived from 3-dehydroquinic acid can be used as a food additive, a cosolvent, and an optical material. Moreover, pyrogallol and chlorogenic acid synthesized via an artificial pathway derived from the shikimate pathway are widely used in various fields. There is a growing demand for these intermediates and their derivatives.

As shown in Table 1, metabolic engineering of the shikimate pathway and its branching pathways is currently mainly performed in *E. coli* and sometimes in *C. glutamicum* and *B. subtilis,* because non-model microbes are expensive, and the development of new genetic tools is time-consuming. The main metabolic engineering methods include: (1) blocking the downstream pathway of the target product to increase product accumulation; (2) modification of the CCM pathway by modifying the PTS and or overexpressing or inactivating related genes to increase the supply of the starting substrates, PEP and E4P pathways; (3) reducing by-product formation to enhance product yield and titer; (4) overexpressing or inactivating key enzyme-encoding genes to relieve feedback inhibition; (5) introducing heterogenic genes to expand metabolic networks, and (6) performing modularization transformation. To date, classical metabolic engineering methods have achieved good results; for instance, the construction of the recombinant *E. coli* SP1.1 PTS/pSC6.090B strain by Chanderan et al. resulted in an SA titer of 87 g/L with a yield of 0.36 mol/mol [46]. The use of classical metabolic engineering strategies to produce shikimic acid has been carried out for decades. Although the titer and purity of SA has improved step by step, there is still a certain gap to replace the traditional SA production method in industries. The limitations of locally modifying synthetic pathways are also limited by editing tools. For example, the gene editing method based on homologous recombination adopted in the traditional metabolic engineering strategy in this paper has the defects of a long cycle and low recombination efficiency. More interestingly, synthetic biology technology has also made good progress in metabolic engineering, thereby showing prospects in solving the problem of flux imbalance that occurs due to classical metabolic engineering, and has now become a research hotspot. This paper mainly investigates the application of classical metabolic engineering in the production of shikimic acid in different microorganisms. It is found that the traditional metabolic engineering mainly strengthens the related pathways of product synthesis but does not examine the cell metabolic network from a global point of view: that is, it does not explore the effects of more deep metabolism and physiological processes on product synthesis in cells, and so traditional metabolic engineering often cannot take into account cell growth and synthesis. It is difficult to further improve the production capacity of cells. Then, from the analysis of the research results using synthetic biology technology, the researchers pay more attention to the overall control of cells, through the comparison of histological data and the analysis of bioinformatics data, such as AlphaFold, to find more potential and effective targets, and then make breakthroughs in the pathway of material transport or in the problem of cell growth and product synthesis. Unlike the traditional top-down process that focuses on optimizing endogenous pathways or reconstructing heterologous natural pathways to produce metabolites, it is possible for synthetic biology to design new specific metabolic pathways for the desired compounds. So far, synthetic biology technology has played an important role in regulation strategies, such as quorum sensing, temperature-sensitive dynamic systems, photogenetic dynamic systems, metabolite response circuits, and so on [63]. It is also worth mentioning that the new genetic tools involved in synthetic biology will greatly shorten the modification cycle and be more economical.

Metabolic engineering for the synthesis of complex compounds with high value, via catalysis of simple compounds has rapidly developed. In recent years, a strategy for modular metabolic network regulation, multivariate modular metabolic engineering (MMME), was developed by metabolic engineering researchers [59]. This strategy requires a smaller number of modular combinations to optimize metabolic pathways on a large scale, which is more systematic than combinatorial engineering and could avoid the high throughput screening process [99]. Currently, MMME, as an important strategy for strain optimization in metabolic engineering, has been successfully applied in different prokaryotic and eukaryotic strains [100,101]. However, with an increase in the complexity of the structure of the target products, the possibility of a series of problems, such as an imbalance in the host biological metabolism caused by the construction of a multi-gene ab-initio synthesis pathway and the toxic effects of intermediates on host cells, has also increased [102]. It is reported that the synthetic scaffold strategy, as a complementary or parallel method, provides modular and highly flexible assembly tools, and has been widely used in metabolic pathways. Post-translational co-localization pathway enzymes are used to regulate the stoichiometric number, distance and spatial orientation of these pathways to improve metabolic flux and can be flexibly applied to multiple enzyme-catalyzed metabolic pathways [103]. Additionally, the recent CRISPR-Cas9 gene editing technology, which has attracted much attention in the field of cell and molecular cloning, can more efficiently modify the genome and edit DNA fragments, both targeted and at multiple sites [104]. In addition, CRISPR technology can effectively regulate transcriptional intensity without heterologous promoters, so as to regulate gene expression in the genome itself [52]. It can optimize the SA pathway and its branches better, providing the possibility to build a larger and more complex module metabolic network in the future.

Furthermore, a biosensor is a kind of instrument that converts the concentration of bioactive substances into electrical signals. It has the characteristics of signal amplification, high accuracy, high sensitivity, and strong specificity. It has been used for the dynamic regulation of metabolic pathways, real-time monitoring of metabolites, high-throughput screening of engineering bacteria, etc., which makes people more effectively realize the optimal combination of various genetic modules. Recently, a novel SA biosensor based on the LysR-type transcriptional regulator (ShiR) of *C. glutamicum* has been constructed. This biosensor can specifically respond to intracellular SA concentration and detect extracellular SA when co-expressed with the *shiA* gene, thus achieving high-throughput screening of *C. glutamicum* with high SA production [105]. This means that we will have a better understanding of the metabolic flux of bacteria and greatly shorten the time of strain screening. More unexpectedly, Niu et al. invented a method combining atmospheric and room temperature plasma mutation guided by biosensor and genome recombination to construct *E. coli* SA-GS-2-7, which can produce 24.64 g/L SA from sucrose in a yield of 1.42 mol/mol [106]. This also suggests that biosensors are important in future metabolic studies.

It is a great idea to produce as many products as possible through bioengineering, but according to the findings of this paper, the limitations of classical metabolic engineering have greatly hindered this process. However, metabolic engineering has been constantly evolving during the past twenty years. With the combination of synthetic biology and systems biology, the concept of systems metabolic engineering has also emerged spontaneously [107]. Reshaping biosynthetic pathways in chassis organisms to achieve optimal productivity must be aided by omics data, computer-aided modeling, and efficient and precise editing tools. Organism is a very complex open system. The improvement of artificial intelligence tools and major biological databases will help us better understand the laws of life, so as to propose better metabolic strategies, continuously enrich the content of systematic metabolic engineering, and develop more advantages of strains that produce higher-value compounds.

## Figures and Tables

**Figure 1 molecules-27-04779-f001:**
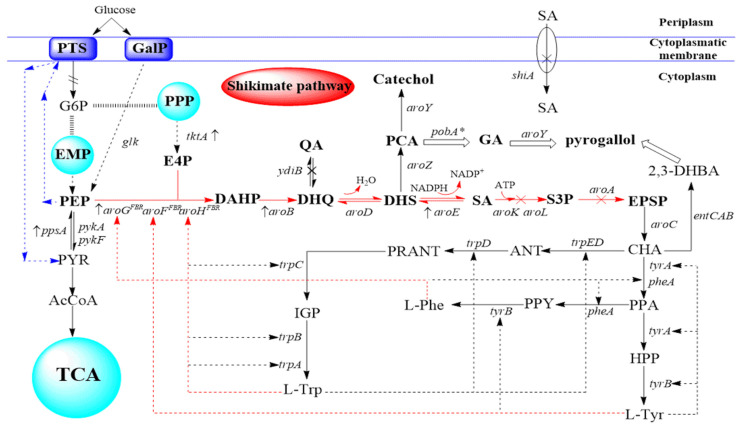
PTS system, central carbon metabolism and shikimate pathway. Solid line arrow, reaction direction; Dash line arrow, feedback inhibition; Empty arrow, artificial biosynthetic pathway; ×, knockout; ↑, Overexpression. 2,3-DHBA, 2,3-dihydroxybenzoic acid; ANT, anthranilic acid; CHA, chorismate; DAHP, 3-deoxy-D-arabino-heptulosonate-7-phosphate; DHQ, 3-dehydroquinic acid; DHS, 3-dehydroshikimic acid; E4P, erythrose-4-phosphate; EMP, Embden-Meyerhof-Parnas pathway; EPSP, 5-enolpyruvylshikimate-3-phosphate; G6P, Glucose 6-phosphat; HPP, 4-hydroxyphenylpyruvate; IGP, indole 3-glycerolphosphate; L-Phe, L-phenylalanine; L-Ser, L-serine; L-Trp, L-tryptophan; L-Tyr, L-tyrosine; PEP, phosphoenolpyruvate; PPA, phenyl-pyruvic acid; PPP, phosphopentose pathway; PPY, phenylpyruvate; PRANT, N-(5-ribose phosphate)-aminobenzoic acid; PRE, prephenic acid; PRPP, 5-ribose phosphate-1-pyrophosphate; PYR, pyruvate; QA, quinic acid; S3P, shikimate-3-phosphate; SA, shikimic acid; TCA, tricarboxylic acid cycle; Genes and coded enzymes: *aroA*, EPSP synthase; *aroB*, DHQ synthase; *aroC*, chorismate synthase; *aroD*, DHQ dehydratase; *aroE*, shikimate dehydrogenase; *aroG*, *aroF*, *aroH*, DAHP synthase isoenzyme genes; *aroK*, shikimate kinase I; *aroL*, shikimate kinase II; *entC,* isochorismate synthase; *entB,* isochorismatase; *entA,* 2,3-dihydro-2,3-DHBA dehydrogenase; *ppsA*, phosphoenolpyruvate synthase; *pykA*, pyruvate kinase II; *pykF*, pyruvate kinase I; *tktA*, transketolase I.

**Figure 2 molecules-27-04779-f002:**
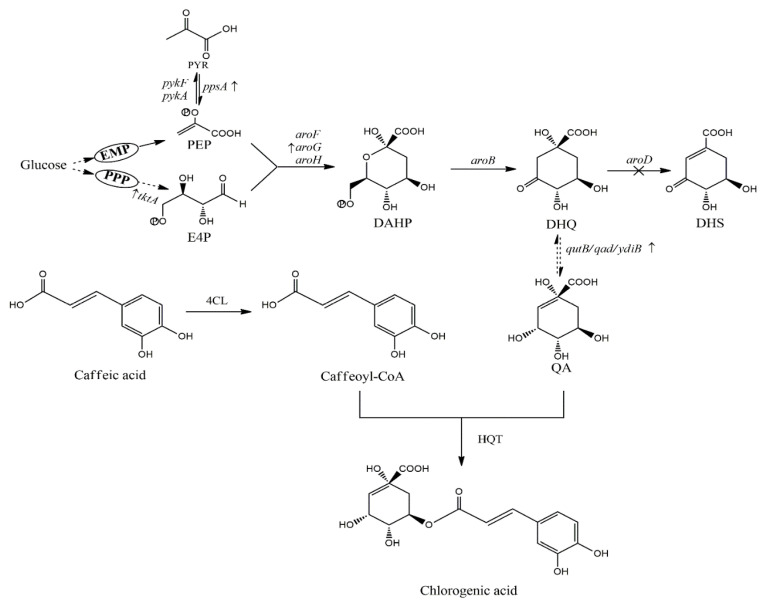
Shikimate pathway and quinic acid synthesis pathway. ×, knockout; ↑, Overexpression; 4CL, 4-coumaric acid: CoA ligase; DAHP, 3-deoxy-D-arabino-heptulosonate-7-phosphate; DHQ, 3-dehydroquinic acid; DHS, 3-dehydroshikimic acid; E4P, erythrose-4-phosphate; EMP, Embden-Meyerhof-Parnas pathway; HQT, hydroxycinnamoyl transferase; PEP, phosphoenolpyruvate; PPP, phosphopentose pathway; PYR, pyruvate; QA, quinic acid; Genes and coded enzymes: *aroB*, DHQ synthase; *aroD*, DHQ dehydratase; *aroF*, DAHP synthase (L-Tyr); *aroG*, DAHP synthase (L-Phe); *aroH*, DAHP synthase (L-Trp); *ppsA*, PEP synthetase; *pykA*, *pykF*, pyruvate kinase II and pyruvate kinase I, respectively; *qutB/qad*, quinic acid dehydrogenase; *tktA*, transketolase I; *ydiB*, quinic/shikimate dehydrogenase.

**Figure 3 molecules-27-04779-f003:**
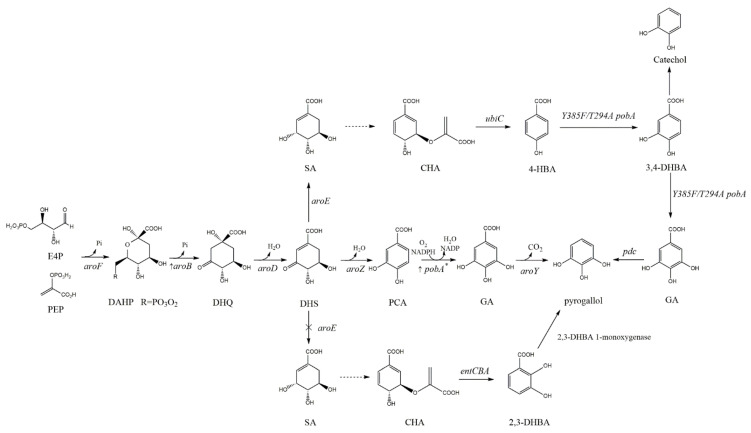
Shikimic acid pathway and gallic acid and pyrogallol synthesis pathway. Dash line arrow, polystep reaction. 4-HBA, 4-hydroxybenzoic acid; GA, gallic acid; PCA, protocatechuic acid. Genes and coded enzymes: *aroB*, DHQ synthase; *aroD*, DHQ dehydratase; *aroF*, DAHP synthase; *aroY*, PCA decarboxylase; *aroZ*, DHS dehydratase; *pobA**, p-hydroxybenzoate hydroxylase; *pdc*, 3,4-dihydroxybenzoic acid decarboxylase; *ubiC*, native chorismate pyruvate lyase; *Y385F/T294A pobA*, p-hydroxybenzoate hydroxylase with Y385F and T294A mutations.

**Figure 4 molecules-27-04779-f004:**
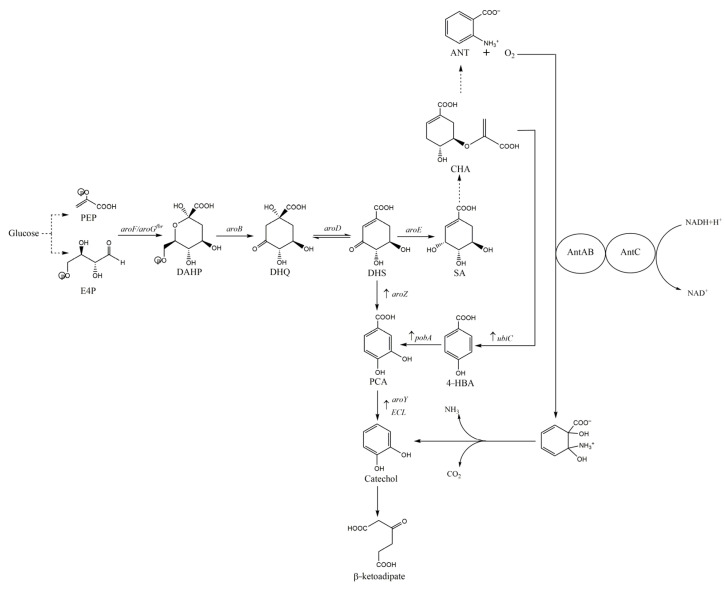
Metabolic pathways related to catechol biosynthesis in *E. coli*. Arrows with dashed lines indicate more than one enzymatic reaction. Abbreviations: ANT, anthranilate; AntAB, terminal oxygenase component; AntC, reductase component of anthranilate 1,2-dioxygenase; DAHP, 3-deoxy-d-arabino-heptulosonate-7-phosphate; DHQ, 3-dehydroquinicacid; DHS, 3-dehydroshikimicacid; E4P, erythrose-4-phosphate; PCA, protocatechuate; PEP, phosphoenolpyruvate; SA, shikimic acid; Genes and coded enzymes: *aroB*, DHQ synthase; *aroD*, DHQ dehydratase; *aroE*, shikimate dehydrogenase; *aroF*, DAHP synthase; *aroG^fbr^*, feedback inhibition resistant DAHP synthase; *ECL,* PCA decarboxylase; *trpED*, anthranilate synthase-phosphoribosyl transferase complex.

**Table 1 molecules-27-04779-t001:** Effects of different metabolic engineering strategies on products.

Strains-Production	Relevant Characteristics	Culture Method	Titer (g/L)	Yield (mol/mol)	Productivity (g/L/h)	Source or Reference
*E. coli* W3110.shik1-SA	∆*aroL*, ↑*aroG^FBR^,* ↑*aroF^FBR^,*	Batch culture	ns	0.059	ns	[36]
*B. megaterium* MTCC428-SA	∆*aroK*	Batch culture	6	ns	ns	[38]
*E. coli* DHPYAAS-T7-SA	∆*ptsHIcrr*, ∆*aroL*, ∆*ydiB*, ↑*aroE*, ↑*aroB*, ↑*glk*, ↑*tktA*, ↑*aroF^fb^*^r^, ∆*aroK*	Shake flask	1.85	0.426	ns	[37]
*C. Glutamicum*-SA	∆*qsuD*, ∆*qsuB*, ∆*aroK*, ∆*ptsH*, and ∆*hdpA*, ↑*tkt,* ↑*tal,* ↑*aroG,* ↑*aroBDE,* ↑*IolT1,* ↑*glk1,* ↑*glk2,* ↑*ppsk),* ↑*gapA*	Fed-Batch	141	0.510	2.938	[51]
*B. subtilis* DE1-SA	∆*aroA*, ↑*aroE130,* ↑*aroD*	Shake flask	1.1	ns	ns	[35]
SK4/*rpsM*-SA	∆*aroL*, ∆*aroK*, ↑*aroB,* ↑*aroG*,* ↑*ppsA,* ↑*tktA,* ColE1 ori, km^r^, *rpsM*, *aroK*	Shake flask	1.74	ns	ns	[39]
SK5/pSK6-SA	∆*aroL*, ∆*aroK*, ∆*ydiB*, ∆*ppc*,∆*ldhA*, ColE1 ori, *aroB, tktA, aroG*,* DHQ/SDH, *rpsM*, *aroK*, Spe^r^	Shake flask	5.33	ns	ns	[39]
*E. coli* SP1.1/pKD12.112-SA	*serA*::*aroB*, *aroL478*::Tn10 *aroK17*::Cm^R^, ↑*aroF^FBR^*, ↑Ptac*aroE*	Fed-Batch	38	0.12	ns	[22]
*E. coli* SP1.1/pKD12.138-SA	*serA*::*aroB*, *aroL478*::Tn10, *aroK17*::Cm^R^, ↑*tktA*	Fed-Batch	52	0.18	0.87	[13]
*E. coli* JB4/pJB5.291-SA	∆*aroD*, ∆*aroE*, ↑*aroF^FBR^,* ↑*tktA,* ↑*aroD·E*	Fed-Batch	34	0.15	0.52	[13]
*E. coli* PB12.SA22 *ydiB*-SA	∆*ydiB*, ∆*aroL*, ∆*aroK*, ↑*pJLB aroG^fbr^*, ↑*tktA,* ↑*aroB,* ↑*aroE*	Batch culture	8.2	0.24	ns	[42]
*E. coli* SP1.1/pKD12.138-SA	*serA*::*aroB*, *aroL478*::Tn10 *aroK17*::Cm^R^, ↑*tktA*	Fed-Batch	35	0.19	0.73	[13]
*E. coli* SP1.1 PTS/pSC6.090B-SA	∆*aroL*, ∆*aroK*, ↑*aroF^Fbr^*, ↑*aroE*, ↑*glf*, ↑*glk,* ↑*tktA*	Fed-Batch	87	0.36	5.2	[46]
*E. coli* SA 110-SA	↑*aroG^fbr^*, ↑*tktA,* ↑*aroB,* ↑*aroE*, ∆*aroL*, ∆*aroK*, *↓**pps,* *↓**csrB*	Shake flask	1.34	0.21	ns	[48]
*E. coli* SA 116-SA	↑*aroG^fbr^*, ↑*tktA,* ↑*aroB,* ↑*aroE*, ↑*pntAB,* ↑*nadK,* ∆*aroL*, ∆*aroK*, *↓**pps,* *↓**csrB*	Shake flask	3.12	0.33	ns	[48]
*E. coli* PB12.SA22-SA	↑*aroG^FBR^*, ↑*tktA*, ↑*aroB*, ↑*aroE*, ∆*aroL*, ∆*aroK*	Shake flask	7	0.29	ns	[43]
*C. glutamicum* RES167Δ*aroK*-SA	∆*aroK*, ↑RiboJ, GFP, (*aroG^MU^, aroB^MU^, aroD^MU^, aroE^MU^*) with various RBS	Fed-Batch	11.40	ns	ns	[60]
*C. glutamicum* TD -SA	↑*tkt,* ∆*pyk*	Shake flask	7.76	ns	ns	[54]
*B. subtilis* BSSA/Ω*pyk*::pSA*pyk*-SA	↑*aroD*, ↑*aroA*, ∆*pyk*	Shake flask	3.46	ns	ns	[56]
*E coli* B0013 SA5/pTH-*aroG^fbr^-ppsA-tktA*-SA	∆*aroL*, ∆*aroK*, ∆*ydiB*	Shake flask	14.6	0.29	ns	[61]
*E. coli* HJS203	HJS200, Amp^r^, Cm^r^, p15A-P_rpsL_-*tetR*-LAA-P_tet_-*aroG^fbr^*-P_P9_-*tktA*-P_P9_-*aroB*^opt^-*aroE*, pPMB1-P_rpsA P1_-*aroK*-DAS+4	Batch culture	14.33	0.23	ns	[62]
*E. coli* S8	*E. coli* MG1655, Δ*ptsHIcrr*::*Zmglf*, Δ*aroL*::*tktA*, Δ*aroK*, P_J23119_-GB, Δ*aroK*::*rpE*, Δ*yidB*, carrying pJ01-TetR(166)-TEVp(118), NX3	Fed-batch	76		1.05	[63]
*E. coli* AB2848*aroD*/pKD136/pTW8090A-QA	∆*aroD*, ↑*qad* (from *Klebsiella pneumoniae*), ↑*tktA,* ↑*aroF*, ↑*aroB*	Shake flask	10.7	ns	0.45	[10]
*E. coli* strain B-101-QA	∆*aroD*, ↑*NtHQT,* ↑*4CL,* ↑*ydiB*	Shake flask	0.45	ns	0.19	[72]
*E. coli* AB2834/pKD136/pKD8.243A-GA	↑*aroE*, ↑*aroB*, ↑*aroZ*(from *K. pneumoniae*), ↑*aroY* (from *K. pneumoniae*), ↑*catA* (from *Acinetobacter calcoaceticus*)	Shake flask	3.75	0.33	ns	[92]
*E. coli* KL7-GA	∆*aroE*, ↑*aroG^fbr^*, ↑*aroZ,* ↑*pobA*,* ↑*aroB*	Fed-Batch	20	0.12	0.42	[84]
strain *Y385F/T294A*-GA	↑ *pobA**	Shake flask	1.15	ns	ns	[85]
*E. coli* JM12-pyrogallol	pCS-*entCBA*-APTA and pZE-nahG^opt^	Shake flask	1.04	ns	ns	[82]
RHY5-pyrogallol	*aroL, ppsA, tktA, aroG^fbr^* from *E. coli, ubiC* from *E. coli, two operons, PLlacO1-cipA-Y385F/T294A pobA* and *PLlacO1-pdc* and *PLlacO1-PDC*	Shake flask	0.08	ns	ns	[86]
W3110 *trpD9923/*pJLB*aroG^fbr^**tktA/*pTrc-*ant3*-Catechol	Δ*ptsH*, *ptsI*, ↑*antABC*, ↑*aroG^fbr^*, ↑*tktA*, ↑*aroF*, ↑*aroB*,	Fed-Batch	4.47	0.16	ns	[21]
*E. coli* WN1/pWL1.290A-Catechol	Δ*aroE*, ↑*tktA*, ↑*aroB,* ↑*aroZ*, ↑*aroF^fbr^*, ↑*aroY*,	Shake flask	8.50	7.00	ns	[94]
*E. coli* N74dpheA pUbic-PobA pECL	Δ*pheA*, ↑*ubiC*,↑*pobA*, ↑*ECL*	Batch culture	0.63	0.06	ns	[96]

ns, not specified. Up arrow, increased gene expression. Down arrow, decreased gene expression.

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
