# Peer review of "Metabolic Engineering of Shikimic Acid Biosynthesis Pathway for the Production of Shikimic Acid and Its Branched Products in Microorganisms: Advances and Prospects"

_molecules, 2022, doi:10.3390/molecules27154779_

Round 1

Reviewer 1 Report

To authors

The work of Wu et al., is a well written contribution to the field of the Shikimate pathway and its derivatives. Though the main literature is reviewed in detail, this reviewer believes the manuscript will benefit from a deeper analysis/discussion.

Following are some point the authors must consider.

1.       In general, there is a well achived description of the different pathway engineering approaches for improving SA production and other derivatives. Yet, some sections are just descriptive; there is no analysis but rather a summary of the different findings achieved so far, e.g., synthetic biology.

2.       Cited literatures is complete, (though in a couple of cases very old). What is missing in the paper is a detailed discussion about differences / analogies found in published works. Please highlight novel insights gained after comparing the available literature about major achievements in the field.

3.        Somehow the paper follows the structure of the Averesch & Kromer´s work (2018), which is not cited.

4.       This reviewer believes, readers will benefit from this review paper if it further includes a dedicated section to summarize the metabolic engineering tools more often used in the different works (presented). Same for the synthetic biology contributions. Further, the manuscript must include a more complete review of literature related to the contribution of omics approches to study the limitations of the shikimate pathway.

5.       Table 1 shows a complete scenario of what is already achieved in SA metabolic engineering efforts. Please consider modify the column named Culture Method. It includes both, mode of operation and type of reactor, which could be separated by groups, so a better critical comparison would be made. The table is rich in content and very informative, so it must be accompanied with a more complete discussion.

6.       In the introduction, the authors state that the manuscript “proposes the application and future prospects of sytnthetic biology, combined with the systemic metabolic engineering strategy in the construction of microbial strains for high-yield SA production”. However, the conclusion section is a modest extension of what is depicted within the body of the manuscript. It lacks wraping up ideas, innovative ideas about major analogies, and missing initiatives. Further, this reviewer suggests, the authors must further discuss the 6 metabolic engineering methods that are cited in the Conclusion section.

Minor comments:

-          Please consider updating some references

-          Please check for references that do not correspond to what is being cited. Also, check for data that refer to old papers

-          Spelling mistakes/typos throughout the document must be carefully amended

L 30: SA has not been previously defined

L 98: Phospho-pentose pathway

L 127 – 127: Incomplete sentence

L 152: Check for typo

L 165: Check for typo

L 171: Check for typo

L 179: Check for typo

L 254: Check for typo

… …

L 638: Check for typo

Reviewer 2 Report

Title: Metabolic engineering of shikimic acid biosynthesis pathway for the production of shikimic acid and its branched products in microorganisms: advances and prospects.

Authors: Wu et al.

Summary: The authors attempted a review of articles focused on genetic engineering of the shikimate and downstream pathways for production of shikimic acid, Chlorogenic acid, quininic acid, gallic acid, pyrogallol and catechol in bacteria. The authors discussed current metabolic engineering targets and approaches to improve SA, QA, chlorogenic acid, GA, pyrogallol and catechol production in bacteria.

Specific comments

1.      The authors did a good job highlighting the usefulness of SA, QA, chlorogenic acid, GA, pyrogallol and catechol.

2.      The authors discussed the successes achieved with recent advances in molecular engineering however; they did not adequately discuss the drawbacks. It is quite difficult to follow a pattern and effectively examine the drawbacks of each strategy due to the fact that they reviewed different genetically modified bacteria with varying production patterns. While this is relevant, the manner in which they are discussed in the manuscript needs to be streamlined for greater clarity. 

3.      Clear separation of metabolic engineering of a specific group of microorganisms (Gram positives or Gram-negatives; e.g., E. coli, B. subtilis or C. glutamicum) would make a better review.

4.      The authors did not effectively discuss the specific application of prospective metabolic engineering approaches to address shortcomings of classical metabolic engineering approaches.

5.      Please rewrite the sentence in line 69-71

6.      Authors should address typographical and grammatical errors in manuscript.

Conclusion: Review article covers scope of title and adequately captured metabolic engineering strategies explored to improve SA, QA, chlorogenic acid, GA, pyrogallol and catechol. Greater effort to streamline the manner in which the different groups of microorganisms have been engineered, the outcomes, shortcomings, and future perspectives will greatly improve the manuscript.

Round 2

Reviewer 1 Report

molecules-1782424-R1

Metabolic engineering of shikimic acid biosynthesis pathway for the production of shikimic acid and its branched products in microorganisms: advances and prospects

Sijia Wu et el.

To authors

Recomendations made by this reviewer have been considered. Thanks. However, there still are some points that need attention.

1.       Comments and analysis on Table 1 (rich) content still are somewhat superficial. Readers would benefit from a deeper comparative analysis.

2.       The paper describes/analysis Metabolic Engineering strategies for SA biosynthetic pathway; however, no single Metabolic engineering tool is even mentioned. The discussion relies on genetic/pathway inprovements but not in the way metabolic targets are identified. Same for omics strategies. These concerns were rised in question # 4.

3.       Many typos/errors remain. Please revise L62, L66, L70, … … … L171, … … L199 … … L363/364, … … L386/387 … …

4.       Resolution of Fig 4 must be improved.
